# Dietary Iron Unlikely to Cause Insulin Resistance in Horses

**DOI:** 10.3390/ani12192510

**Published:** 2022-09-21

**Authors:** Nancy L. McLean, Nerida McGilchrist, Brian D. Nielsen

**Affiliations:** 1The Royal (Dick) School of Veterinary Studies, University of Edinburgh, Easter Bush Campus, Midlothian EH25 9RG, UK; 2Equilize Horse Nutrition Pty Ltd., P.O. Box 11034, Tamworth, NSW 2340, Australia; 3Department of Animal Science, Michigan State University, 474 S. Shaw Lane, East Lansing, MI 48824, USA

**Keywords:** iron, insulin resistance, racehorse, metabolic disease, nutrition, equine, horse

## Abstract

**Simple Summary:**

In the equine diet, iron comes from both roughage and concentrate, as well as often being supplemented with the expectation that it will improve performance and health. This is commonly done in the racehorse industry. To determine iron consumption in this population of horses, a survey of 120 U.S. Thoroughbred trainers, representing 1978 Thoroughbreds from various regions of the U.S., was conducted. Racehorses were fed an average of 3900 mg of iron per day from hay and grain alone. This exceeds the recommendations put forth by the 2007 Horse NRC of 0.8 mg/kg BW or 400 mg for a 500 kg working horse. Supplements increased the daily average intake by an additional 500 mg Fe. Despite some equine nutritionists suggesting excess dietary Fe may be a contributing factor in the development of insulin resistance (IR), there was not one case of IR in any of the trainer’s Thoroughbred horses. Given the excessive iron provided to the horses in this study, it is unlikely dietary iron is an independent causative factor of IR.

**Abstract:**

Racehorses are often supplemented extra iron with the expectation that the iron will improve overall performance and health. A survey of 120 U.S. Thoroughbred trainers, representing 1978 Thoroughbreds from various regions of the U.S., was conducted to determine the average amount of dietary iron fed to Thoroughbred racehorses per day. Survey results indicated racehorses were fed an average of 3900 mg of iron per day from hay and grain alone. This exceeds the 0.8 mg/kg BW or 400 mg for a 500 kg working horse that the NRC 2007 recommends per day. Supplements increased the daily average intake of iron by an additional 500 mg Fe. Some equine nutritionists propose that excess dietary iron may be a causative factor in insulin resistance (IR). However, the occurrence of IR in Thoroughbred racehorses is very rare. This study did not find one confirmed veterinary diagnosis of IR in any of the surveyed trainers’ Thoroughbred horses, whether racing, on a layoff, or retired. Given the iron content in these diets easily exceeds the NRC minimum daily requirements, it seems unlikely that dietary iron is an independent causative factor in IR.

## 1. Introduction

In recent years, dietary iron (Fe) has been linked as a possible cause of insulin resistance (IR) in horses. One of the first studies that correlated Fe level with IR reported an elevation of body Fe in IR horses with uncontrolled mineral intake [1]. As captive rhinoceroses often experience iron overload (IO), another project used horses as a model for Black Rhinoceroses to look for a possible relationship between Fe status and IR [2]. A positive correlation was found between indices of high body Fe stores (serum ferritin concentrations [3]) and insulin insensitivity in a glycemic challenge. However, the results suggested the IR condition increased the incidence of IO disorder, as opposed to the excess Fe acting as the catalyst for the development of the IR. Other studies have not shown a significant linear relationship between iron and hyperinsulinemia. Theelen et al. [4] studied equids with severe iron overload yet there was no evidence of metabolic dysfunction. Pearson and Andreasen [5] fed ponies high amounts of ferric citrate without any signs of iron toxicosis occurring, and LaVoie and Teuscher [6] researched ponies fed high amounts of dietary iron without any findings of metabolic disorder. Despite this, the Nielsen et al. [2], study prompted concerns by some in the horse industry that excess dietary Fe causes IR.

To date, this IO/IR theory has not been fully researched in horses; however, in humans there appears to be some correlation.

Research results suggest that Fe can modify insulin sensitivity by interfering with insulin receptors and intracellular insulin signaling [7]. High Fe stores seem to be predictors of type 2 diabetic onset, while low Fe is deemed protective [8]. This proved to be true across the broad human demographics of pre-menopausal and post-menopausal women, as well as in men [9]. More recently, it was concluded that Fe metabolism plays a role in the development of IR and type 2 diabetes, via the effect that Fe metabolism has on liver function [10], while Zafar et al. [11] also found that non-diabetic offspring of type 2 diabetics had higher serum ferritin and transferrin saturation than non-diabetic offspring of non-diabetics, indicating a genetic component.

Human Metabolic Syndrome and Equine Metabolic Syndrome (EMS) share striking similarities. These include increased body mass index (BMI) or obesity, intra-abdominal or visceral obesity (adiposity in horses), hyperinsulinemia, IR, hypertension, pro-inflammatory state, and altered adipokines (increased leptins/decreased adiponectins in horses) [12].

As a link suggesting IO caused IR was proposed [13], articles blaming excess dietary Fe on IR appeared in horse-interest publications and websites. However, without further research, concerns about excess dietary Fe seem premature. If IO proves to be a causative factor of IR, then dietary and nutritional protocols need to be evaluated accordingly. However, if excess dietary Fe is not causative of IR, then such concerns are unheeded.

According to both the 1989 [14] and 2007 [15] NRC, the minimum Fe requirements for mature horses is 40 mg Fe/kg DM. On a BW basis for a working horse, the requirement is 0.8 mg/kg BW (which works out to 400 mg Fe per day for a 500 kg horse). Further, the maximum tolerable concentration of dietary Fe has been suggested to be 500 mg Fe/kg DM, equivalent to ~5000 mg Fe per day for a 500 kg horse [16].

The practice of providing racehorses with supplemental vitamins and minerals to improve performance has long been practiced. An Australian study reported a group of Thoroughbred racehorses had an average daily Fe intake that exceeded the recommended NRC 1989 [14] amount by over 200% [17]. Further, Richards and Nielsen [18] found that few racehorses receive less than 300% of their daily Fe requirement.

With racehorses being a population of horses commonly supplemented with Fe, this study was done to determine the average daily dietary Fe intake in a population of Thoroughbred racehorses in the United States and to access any link caused by excessive Fe intake and IR. The hypothesis of this study was that Thoroughbred racehorses in the United States are commonly fed high concentrations of dietary Fe, and this practice does not increase the occurrence of IR in this population of horses.

## 2. Materials and Methods

The Human (Research) Ethical Review Committee (HERC) at the University of Edinburgh approved this study (HERC_252-18). Informed consent was obtained from all subjects involved in the study.

This research was completed by the administration of a survey created through Online Survey (formally Bristol Online Survey). The survey was made up of 14 questions (Table 1) that provided multiple choice options to simplify the process and reduce respondent burden [19]. An open text box was provided if the respondent needed to explain an answer or choose an option that was not offered.

The survey was distributed to U.S. licensed Thoroughbred trainers, via an online link, email, phone, and face-to-face interviews at various racetrack locations. Ed Bowen, Past-President of the Jockey Club, assisted this study by sending an online survey link to U.S. licensed trainers from his personal email contact list. The Jockey Club had no influence or affiliation with this survey.

The target list had 275 trainers representing each racing jurisdiction in the 31 states that are licensed to hold Thoroughbred race meets. These trainers represented all classes of racing, from low-level claiming ranks to graded stakes races. Some trainers had retired Thoroughbreds and horses currently recessed from racing (“layups”) that were still under their management either on a farm or at the track. The survey began in early December and had a consistent flow of respondents through April. It should be noted that during the latter days of this survey, U.S. racing was under scrutiny for a high number of fatalities during training/racing at the Santa Anita Racetrack. Response rate did slow, and some trainers seemed hesitant to respond, but with a conversational approach (either by phone or face-to-face interviews), and the assurance of anonymity, responses continued to be recorded.

Face-to-face interviews were completed by mobile application linked to the online survey, as a type of Computer-Assisted Personal Interviewing (CAPI) system or by paper surveys [19]. Once the survey was in progress, results were monitored daily through the online survey’s email updates.

The survey questions covered the average amount of forages and concentrates/grains fed per day, as well as the use of Fe supplements and multivitamins. The occurrence of IR within a stable was also recorded whether by trainer observation (signs/symptoms were listed on the survey in Question 14) and/or by veterinary diagnosis.

Roughages were classified as straight alfalfa, majority alfalfa with a small amount of grass mix (70/30), 50/50 alfalfa/grass mix, and grass mix only. This is representative of roughages available throughout the United States. Grains/concentrates had an open text box since diets are extremely varied between trainers and geographic regions.

To calculate the estimated daily Fe intake from roughages, the average Fe concentrations for mixed grass hay (MMG), alfalfa (Legume), and mixed alfalfa (MML) reported by EquiAnalytical [20] were used, with amounts reported in ppm (mg Fe/kg DM). The average concentrations were 343 mg/kg DM for MMG, 401 mg/kg DM for Legume, and 269 mg/kg DM for MML. Using survey responses for amounts of roughages fed and multiplying those amounts by the average Fe concentrations reported by EquiAnalytical [20] in those types of hay, the estimated daily Fe intake from roughages was determined.

To determine Fe content of concentrates, manufacturers were contacted by phone and email, and the Fe concentrations present in the most current batch for grain and premix concentrates were provided. Multiplying those Fe concentrations by the amount of grain fed by trainers, daily Fe intake from concentrates/grains was determined.

Manufacturer labels and websites were used to determine the concentrations of Fe in various multivitamin and mineral supplements, with the amount provided by trainers as indicated in their survey responses used to determine the amount of supplementary Fe provided beyond the daily amount consumed in the roughage and grains/concentrates.

Data were exported from the online survey into Microsoft Excel. Survey statistics were applied to get simple descriptive analysis of the data. Roughages, grain/concentrates, and supplement amounts were tallied and compared to the NRC 2007 recommendations for daily iron minimum (40 mg Fe/kg DM) and to the NRC 2005 maximum tolerable amounts for working horses (500 mg/kg DM). The main area of interest was finding the average amounts of Fe Thoroughbred racehorses consumed per day and comparing it to the incidence of IR.

## 3. Results

### 3.1. Respondents

The number of respondents to this survey was 120 licensed Thoroughbred trainers representing 1978 Thoroughbred racehorses in 25 states. The survey response rate was 43.6%. The largest Thoroughbred stable that participated had 95 Thoroughbreds in training. The smallest stable was a trainer, with over 60 years of experience, currently training one to two Thoroughbred racehorses per year. Overall, trainers had an average of 16 horses in training.

Illinois had the highest response rate (20), followed by Kentucky (14), Ohio (10), and Florida (8), as well as Arkansas, Michigan, Indiana, and Iowa (7). The remaining states had between one and six respondents.

### 3.2. Roughages

The average amount of hay fed per day was 8.1 kg, similar to what was reported by Southwood et al. [17]. This represented 58% of the total diet on an as fed basis. About two-thirds of the trainers fed an alfalfa mix hay, with over half (53.3%) feeding what was deemed as 50/50 mixed hay (Figure 1). Based upon roughage intake and the average Fe concentrations of the roughages, the daily dietary Fe provided by roughages was calculated to average 2414 mg of Fe.

### 3.3. Grains/Concentrates

The quantity of grain/concentrates provided daily ranged from an estimated 2.3 kg to 9.1 kg as-fed, with the average amount of grain/concentrates fed estimated at 5.8 kg per day, representing 42% of the diet. This is comparable to Thoroughbreds in a study by Southwood et al. [17] but slightly less than the 7.3 kg reported by Richards et al. [21]. Sweet feed was the primary choice for grain/concentrates with 94% of trainers feeding at least some sweet feed, whether in entirety or in part with other grains. This includes trainers that fed a pelleted concentrate mixed with sweet feed to increase palatability. Based upon the amounts provided and the Fe concentration within the concentrates/grains, daily Fe intake provided by this portion of the diet averaged 1476 mg. Before considering supplemental Fe, the daily intake of Fe from roughage and concentrate averaged 3887 mg with a range from a low of 2159 mg to a high of 5821 mg.

### 3.4. Usage of Supplements Containing Iron

According to this survey, 68.4% of trainers (82/120) provided an Fe supplement and 93.3% fed a multivitamin supplement (112/120). Of the multivitamin supplements, 60% (12/20) contained additional Fe. Of those multivitamin supplements that contained Fe, the amount provided in recommended daily serving would range from 190 mg to 1870 mg. Surprisingly, 40% of trainers supplemented Fe with both an Fe supplement and a multivitamin that also contained Fe. The main reason for supplementing Fe, according to the survey, was to prevent anemia (45% or 54/120). A secondary reason was to boost red blood cells for oxygen production to increase performance (18% or 22/120).

The current survey results are similar to findings of Richards et al. [21] who reported that 58.3% of trainers supplemented Fe and 98.6% used some form of dietary supplements. The manufacturer-recommended daily doses of Fe ranged from a minimum of 170 mg to a maximum of 1870 mg. The amount provided daily in an Fe-specific supplement averaged 376 mg. Combined with the feedstuffs, the average intake was 4411 mg Fe daily. While not all multivitamin supplements contained Fe, the provision of those that did would increase the daily Fe intake above that.

### 3.5. Supplements/Vitamins during “Layoffs”

While about 60% of trainers did not provide Fe supplements or multivitamin supplements during stall rest or “layoffs”, or with retired racehorses, nearly 40% did. This finding points to the fact that even when not exercising, Fe intake typically greatly exceeded requirements.

### 3.6. Incidence of Insulin Resistance

Of the 1978 horses represented by 120 trainers from 25 states within the United States, and representing horses both in active training and those being rested or retired, there were no reported cases of IR (0%), either by veterinarian confirmation (Survey Question 13, Table 1) or by signs (Survey Question 14)

## 4. Discussion

While it has been suggested that high dietary Fe is causative in IR [13], this study does not find such a link. Racehorses were chosen as a population for survey due to the commonality of supplementing Fe in an attempt to improve the oxygen-carrying capacity of the blood, and thus improve racing performance. The lowest estimated daily intake of Fe without supplementation was 2159 mg per day. Using the average daily feed intake of 13.9 kg, this would work out to 155 ppm—almost four times the minimum intake (40 ppm) recommended by the 2007 NRC [15]. The highest estimated daily intake of Fe without supplementation was 5821 mg. Again, using an average intake of 13.9 kg, this would work out to over 400 ppm—close to the maximum tolerable amount of 500 ppm recommended by the 2005 NRC [16]. Depending upon whether Fe-containing supplements were fed (which were provided to the majority of horses), it is likely there were subject horses being provided Fe that exceeded the maximum tolerable amount.

This agrees with the work of Richards and Nielsen [18] that examined hay samples submitted for analysis to Equi-Analytical in 2017. Representing 5837 hay samples, all hay types (grass hay, legume hay, mixed mainly grass hay, and mixed mainly legume hay) had an average Fe concentration more than five times that required by the athletic horse. Further, 707 hay samples (12%) contained Fe at or above the maximal tolerable amount set by the 2005 NRC [16]. In contrast, only 81 samples (1.4%) contained Fe at less than 50 ppm, and only 15 (0.3%) samples contained Fe at less than 40 ppm. From their analysis, and based upon previously reported feed intakes, it was projected that few racehorses would receive less than 300% of their daily Fe requirement and likely many would receive substantially greater amounts. Besides confirming that most racehorse diets meet Fe requirements without supplementation, Richards and Nielsen [18] acknowledged the dearth of reported IR in Thoroughbred racehorses despite the high intake of Fe. Granted, it is likely not all horse and pony breeds respond similarly. That being said, the current study confirms that IR is seemingly non-existent in racehorses. While Thoroughbreds are known as a breed with naturally high insulin sensitivity, without a single reported case of IR in 1987 Thoroughbred horses being fed high Fe, it is unlikely that high Fe intake alone is a direct and independent causative factor of IR.

As indicated by Nielsen et al. [2], horses that are IR may have increased Fe stores. Of the horses on that project, all were receiving similar diets and would have similar expected Fe intake. However, only the horses that were IR had elevated serum ferritin concentrations. This suggests that being IR is causative of IO, as opposed to IO being causative of IR. It is, of course, possible that some horses in this study had greater Fe absorption than others and that this may have led to IR. However, in humans it has been shown that even individuals with hemochromatosis, a genetic condition that gives rise to the absorption of excessive amounts of Fe, do not have any higher incidence of IR than human subjects without this condition [22,23]

It should also be kept in mind at this point that serum ferritin is not necessarily indicative of Fe overload as Fe and ferritin can be regulated independently from one another [24]. Instead, serum ferritin is a known marker of inflammation and cellular damage and may in fact not be ‘carrying’ much Fe at all. Thus, in the study of Nielsen et al. [2], what was likely being observed was indeed elevated serum ferritin in response to the inflammation caused by IR, independent of any state of IO as reported in a review by Kell et al. [25].

There are many factors that can contribute to the development of IR in horses, including genetics, management, and nutrition [26,27,28,29]. Overfeeding for the level of activity (resulting in obesity) has been acknowledged as one of the main factors leading to the onset of IR in horses [30,31,32]. In looking at the effects of activity/exercise on IR, there is a recognized correlation between exercise and insulin sensitivity in horses [33,34,35]. As such, it is not surprising that there were no reported cases of racehorses in training having IR. However, despite being given high amounts of dietary Fe, there were also no reports of horses with IR or secondary IO conditions during any layoff/rehabilitation time.

Horse breed appears to be linked to the development of IR. Studies show a strong association in horses and ponies between a certain metabolic phenotype and predisposition to laminitis, with an emphasis on obesity and IR [30,32,36,37,38,39]. While Thoroughbreds do not seem to have the same genetic propensity for IR as pony breeds, there have been some veterinary-diagnosed Thoroughbred cases [36]. The most common causes, or precursors, of IR are usually not seen in Thoroughbred horses. Thoroughbred racehorses in training are typically lean, fit, young, extremely athletic, and active.

By contrast, there is a tendency in some other breeds to favor horses that are overweight. Recently, Munjizun et al. [40] reported a positive correlation between the body condition score and the score received for conformation and appearance in elite large ponies at a national championship horse show, demonstrating judges prefer an animal that is over-conditioned. With the average body condition score being 6.6 +/− 0.6 (on a 1 to 9 scale) for the ponies in that study, it suggests horse show judges are rewarding obesity rather than a healthier body condition. Favoring the appearance of a horse that is overweight is a common problem in the horse industry. Combined with inadequate exercise, the onset of IR can be an expected outcome. Not surprisingly, if kept at a healthier weight and provided sufficient exercise, the onset of IR appears to be rare.

Thus, it is important that horse owners be urged to pay greater attention to the factors we know lead to IR, specifically to keep horses at a healthy body condition and to keep them as active as possible.

This present study has several obvious limitations. As with any survey, results are only as accurate as the information provided by the respondents. While most racehorse trainers have a general idea as to the quantity of feed provided to their horses, in this survey, the trainers provided only their best estimate, and they also only provided a single response that is representative of their entire stable. Providing individual feeding programs for every horse involved with this study would be cumbersome and would likely result in a much lower participation rate than the 43.6% that was achieved. Sex and age were also not reported for the same reason. Further, for the roughage portion of the diet, the mean Fe concentration of similar hay samples analyzed by Equi-Analytical were used [20]. Using mineral analysis performed on hay samples might be more accurate, but only if the same hay source was used for an extended period. Hay fed at racetracks tends to be delivered frequently due to limited ability to store hay and it is recognized that Fe concentrations often vary dramatically with different hay sources. Thus, using the mean values from Equ-Analytical seemed prudent rather than requiring mineral analyses being done on the hay each trainer was currently feeding (again, limiting participation in the study, increasing costs, and placing too much emphasis on the Fe concentration in the batch of hay currently being fed). With these limitations, it should be recognized that the Fe intake presented in this study should only be used as an estimate as to total Fe consumption. Additionally, soil Fe intake for horses with access to grazing, and the horse’s drinking water Fe content, were not taken into account, and this has the ability to significantly increase daily overall Fe intake. Further, no attempt was made to quantify the form in which the Fe was presented to the horses (ferrous versus ferric—with obvious differences in absorption rate between the two).

Despite these limitations, with confidence it can be concluded that horses in this study were being fed Fe well above their requirements [15] and, in some cases, probably above the accepted maximum tolerable limit [16]. Recognizing such, the most dramatic finding that supports the hypothesis of this study is that not a single trainer reported any veterinarian-confirmed cases of IR, nor even a single sign of IR, in any of the 1978 horses. This confirms that dietary Fe, provided well above established requirements, is not an independent causative factor of IR.

## 5. Conclusions

This study demonstrated that greatly exceeding recommended dietary amounts of Fe did not result in a single reported case of IR in 1978 Thoroughbreds, confirming the hypothesis that, in isolation, high dietary Fe does not increase the incidence of IR. While there are no reported benefits of feeding high amounts of Fe, there are also no reported negative health consequences of feeding extra Fe (within the range discussed in this paper) in horses. The normal diet a racehorse receives, without supplementation, typically exceeds the required amounts of Fe by several fold, and there is no justification for supplementing more Fe. However, there also is no good justification for trying to select feedstuffs for low Fe concentration. It is not known whether IO is associated as a part of a multi-factorial disease cascade with IR in horses. Until research can determine this, greater emphasis needs to be placed on maintaining a healthy bodyweight for horses and ensuring adequate exercise to maintain insulin sensitivity.

## Figures and Tables

**Figure 1 animals-12-02510-f001:**
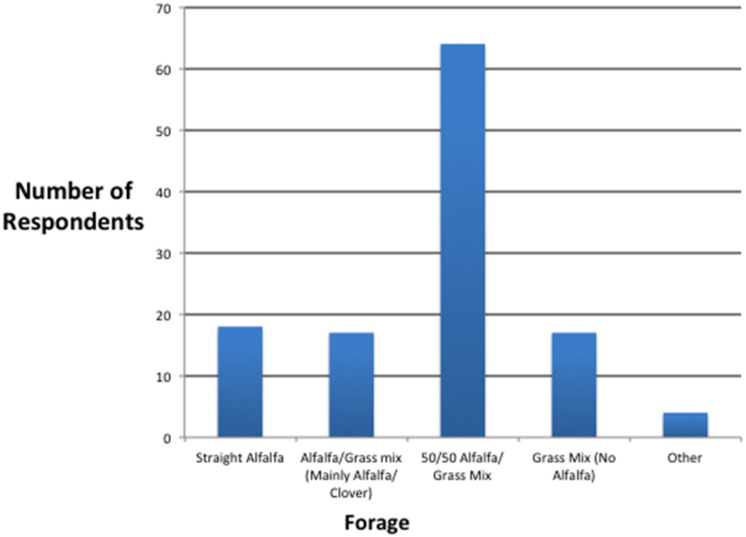
Number of respondents feeding various hay types.

**Table 1 animals-12-02510-t001:** Survey questions provided to Thoroughbred racehorse trainers.

**1. By clicking on the “Agree” tab or answering the survey questions, you agree to the Privacy/Consent Statement.** **2. Please list your racetrack location:** **(Online Survey had a drop-down menu)** **3. What type of hay do you generally feed your stable of Thoroughbreds?** Straight Alfalfa Alfalfa/Grass mix; (Mainly Alfalfa/Clover) 50/50 Alfalfa/Grass Mix Grass Mix (No Alfalfa) Other: (Please list) **4. How much hay do you feed on average per day/per horse? (Including hay substitutes, such as, cubes, pellets, shreds etc.)** Less than 15 lbs 15 lbs (6–8 flakes) 20 lbs (9–10 flakes) More than 20 lbs Free Choice (unlimited) **5. What type/brand of grain concentrates do you feed?** ______________________________________________________________________. **6. How much grain concentrates do you, on average, feed per horse/per day?** Less than 5 lbs 2 Scoops (5 lbs) 4 Scoops (10 lbs) 6 Scoops (15 lbs) 8 Scoops (20 lbs) More than 20 lbs **7. Do you give an iron supplement to your Thoroughbreds in race training?** No, I do not feed an iron supplement Red Cell Perktone Lixotinic Iron Power Other: ______________________. **8. How much iron supplement do you feed per horse/per day?** Recommended dose. A handful per horse More than recommended dose 1–2 Streams (squirts) from a dispenser 3 or more Streams (squirts) from a dispenser **9. What is your main reason for giving an iron supplement?** Prevent anemia EIPH (Exercise-Induced Pulmonary Hemorrhage) Red cell booster for oxygen production Immune builder I have always fed extra iron Iron is not supplemented Other: __________________________________. **10. Do you feed a multivitamin?** Yes, I use Platinum Performance Yes, I use dac Racing Formula No, I do not feed a multivitamin. Yes, I use Farnum VitaPlus Yes, I use Accel Vitaflex Yes, I use dac Total Performance Other: ______________________________. **11. What dosage of multivitamin do you feed per day?** Recommended dose with measuring scoop provided. A handful per horse. Less than recommended dose. More than recommended dose. No multivitamins fed. **12. Do you continue to give supplements to horses that are on stall rest or extended layoffs (walking only)?** Yes No Not applicable/I do not supplement **13. Have any of your Thoroughbred horses, whether in race training, on lay off, or retired, ever been diagnosed by a veterinarian with Insulin Resistance (IR)?** Yes No **13. (a) If yes, what Career Stage was the IR Thoroughbred?** In race training Lay off Retired **14. To your knowledge, have any of your thoroughbreds, whether racing, laid off, or retired, exhibited signs of IR (without veterinary oversight) (such as fat deposits on the shoulders, rump and (cresty) neck, excessive sweating, and slow to shed winter coat)?** Yes No **15. How many Thoroughbreds are currently in your care? ________________.** Thank you for your participation. All data management in this survey is compliant with the General Data Protection Regulations (2018) and the University of Edinburgh Data Protection procedures.

## Data Availability

The data presented in this study are available on request from the corresponding author.

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
