# Peer review of "Dietary Iron Unlikely to Cause Insulin Resistance in Horses"

_animals, 2022, doi:10.3390/ani12192510_

Round 1
Reviewer 1 Report
I have no issues with the content of the manuscript, nor the methods or conclusions drawn. I look forward to the publication of this paper. All my questions were actually addressed by the authors in the discussion section. Suggested edits: page 6, section 3.3 Grains/concentrates, first sentence - "The quantify of grain/concentrates..." should be quantity? page 9, 3rd paragraph (beginning with "This present study..."), 5th sentence - "analysed" should be analyzed.
Author Response
Thank you for reviewing this manuscript and suggesting the following edits listed below.
Quantify has been edited to quantity (page 6) and analysed amended to analyzed (page 9). I appreciate your time.
Nancy McLean
Reviewer 2 Report
If possible with existing literature, elaborate on the introduction. Some of Dr. Kellon's works have brought up blackened livers and theories on the why behind IR/IO theory with high iron. May be worthwhile to include info beyond the rhino paper.
How did iron content in water factor in?
Some may associate iron overload as being an aged horse issue, although it could be assumed that most of the thoroughbreds reported on would be young - any info on age span of the population would be helpful if available.
Author Response
Thank you very much for your suggestions. I appreciate your time spent reviewing this manuscript.
Existing literature has been added to the introduction. (Please see the bottom of page 1 to the top of page 2. References have also been amended.)
Water Fe analysis was not performed. Per your recommendation this was added as a limiting factor of the study. (Please see page 9, third paragraph.)
Reporting on sex and age of the horses in the study was not a survey request. The study included females and both intact and gelded males. Thoroughbreds of racing age were the primary population in this project. Retired horses and those serving as "pony" horses were also included. In response to your review, this information was added to the limitations of the study. (See page 9.)
Thank you for your insight.
Reviewer 3 Report
Dear Authors, please correct this word on page two!

Author Response
Thank you for your review and suggested edit. The abbreviation "BM" has been changed to "BW" on page one and two.
I appreciate the time you spent reading this manuscript.
Reviewer 4 Report
Dietary Iron Unlikely to Cause Insulin Resistance in Horses by McLean et al. is a manuscript that I have thoroughly studied. The investigation into iron intake and insulin resistance is intriguing.
The findings of the survey conducted among racehorse breeders were assessed in this study. The impact of increased dietary Fe concentration on insulin resistance was examined.
The work is actually thoughtfully prepared and written. However, if the authors include some parameters, the article's worth will grow. For instance, the wording should ideally include the breed, sex, and age of the horses.
The abstract section is sufficient it adequately describes the study's objectives, methodology, and findings. The introduction is clear and understandable and the material and method are well written.
References, a discussion, and a conclusion are sufficient. Additionally, the language is highly fluid and simple to understand. I advise minor revision.
Author Response
Thank you for your review and suggestions on the parameters of the study. While it would be interesting to know the age and sex of the horses in the study, such information was not collected. The primary horse population involved in this study were Thoroughbred racehorses of racing age, though the study also included retired racehorses or horses serving as pony horses who may have been older. The study population included females and both intact and gelded males. Considering your recommendations, the fact that we did not gather sex and age specifics has been added to the limitations of the study. (See page 9.)
Thank you for your time reviewing this manuscript and your suggestions.